# Kit Ligand and Kit receptor tyrosine kinase sustain synaptic inhibition of Purkinje cells

Tariq Zaman[1], Daniel Vogt[1], Jeremy Prokop[2], Qusai Abdulkhaliq Alsabia[1], Gabriel Simms[1], April Stafford[1], Bryan W Luikart[3], Michael R Williams[1]*

[1]Department of Pediatrics & Human Development, College of Human Medicine, Michigan State University, East Lansing, United States; [2]Office of Research, Corewell Health, Grand Rapids, United States; [3]Department of Molecular and Systems Biology, Geisel School of Medicine at Dartmouth College, Hanover, United States

**Abstract** The cell-type-specific expression of ligand/receptor and cell-adhesion molecules is a fundamental mechanism through which neurons regulate connectivity. Here, we determine a functional relevance of the long-established mutually exclusive expression of the receptor tyrosine kinase Kit and the trans-membrane protein Kit Ligand by discrete populations of neurons in the mammalian brain. Kit is enriched in molecular layer interneurons (MLIs) of the cerebellar cortex (i.e., stellate and basket cells), while cerebellar Kit Ligand is selectively expressed by a target of their inhibition, Purkinje cells (PCs). By in vivo genetic manipulation spanning embryonic development through adulthood, we demonstrate that PC Kit Ligand and MLI Kit are required for, and capable of driving changes in, the inhibition of PCs. Collectively, these works in mice demonstrate that the Kit Ligand/Kit receptor dyad sustains mammalian central synapse function and suggest a rationale for the affiliation of Kit mutation with neurodevelopmental disorders.

*For correspondence:
will3434@msu.edu

Competing interest: The authors declare that no competing interests exist.

## eLife assessment

This **valuable** study from Zaman et al. demonstrates that the cKit-Kit ligand complex is necessary for the formation and/or maintenance of molecular layer interneuron synapses in cerebellar Purkinje cells. The evidence presented is **convincing**; in particular, the use of cell-type-specific knockout of cKit in molecular layer interneurons and knockout of Kit ligand in Purkinje cells provides robust evidence. This work will be of particular relevance to those interested in inhibitory synapse formation or the role of inhibition in Purkinje cell behavior.

## Introduction

The proto-oncogene receptor tyrosine kinase Kit (aliases c-Kit, CD117; human gene symbol *KIT*, mouse gene symbol *Kit*) is an evolutionarily conserved, loss-of-function intolerant, gene that is enriched in specific neuron populations and tentatively associated with rare cases of neurological dysfunction (pLI = 0.98 and LOEUF of 0.17) (*Figure 1—figure supplement 1A and B*; *Chen et al., 2022*; *Bernstein et al., 1990*). It is perhaps unsurprising that loss-of-function *Kit* mutations in humans are rare; *Kit* is highly pleiotropic, supporting diverse cell populations, including melanocytes, hematopoietic stem/progenitor cells, germ cells, masts cell, and interstitial cells of Cajal (*Yarden et al., 1987*). Thus, *Kit* mutations can result in diverse conditions such as hypo- or hyperpigmentation or melanoma, anemia or leukemia, infertility, mastocytosis, and impaired gut motility or gastrointestinal tumors (Kit biology reviewed *Bernstein et al., 1990*; *Lennartsson and Rönnstrand, 2012*).

The activation of Kit is stimulated by Kit Ligand (KL) (human gene symbol *KITLG*, mouse gene symbol *Kitl*, herein KL), a single-pass transmembrane protein having an extracellular active domain that induces Kit dimerization and kinase activity. Across several decades and species, it has been reported that KL and Kit are mutually expressed in discrete neuron populations known to be connected; it has thus been hypothesized that the expression pattern of KL-Kit may reflect a role in connectivity (*Keshet et al., 1991*; *Motro et al., 1991*; *Hirota et al., 1992*; *Manova et al., 1992*). Consistent with such a function, case reports have implicated inactivating *Kit* mutations with disorders of the central nervous system (*Telfer et al., 1971*; *Finucane et al., 1991*; *Funderburk and Crandall, 1974*; *Lacassie et al., 1977*; *Kilsby et al., 2013*; *Figure 1—figure supplement 1C*). Clinical phenotypes include developmental delay, ataxia, hypotonia, intellectual disability, deafness, and autism spectrum disorder. White matter abnormalities have been noted, and limited evidence suggests KL-Kit may indeed regulate axon outgrowth. KL or Kit mutation or pharmacological manipulation alters outgrowth of spinal commissural axons in vitro (*Gore et al., 2008*), KL stimulates cortical neurite outgrowth in vitro (*Su et al., 2013*), and Kit reduction in cortex of developing rats or mice delays axon extension (*Guijarro et al., 2013*). While these reports provide anatomical/morphological data suggesting that KL-Kit influences connectivity, these studies did not address the consequences to neuronal physiology or synaptic connectivity. Functional studies of KL-Kit in synaptic physiology have likely been hampered by pleiotropy and organismal or cell death. Severe global KL-Kit hypomorphs exhibit fatal anemia, and developmental Kit depletion or inhibition in the cerebrum has resulted in the death of neuronal progenitors or nascent neurons (*Guijarro et al., 2013*; *Mashayekhi and Gholizadeh, 2011*; *Aoki et al., 2017*). Therefore, whether KL-Kit is necessary for synaptic function has remained largely unstudied. Here, we address this gap.

In mice, rats, and humans, Kit is abundantly expressed by molecular layer interneurons (MLIs) of the cerebellar cortex (*Motro et al., 1991*; *Hirota et al., 1992*; *Manova et al., 1992*; *Kim et al., 2003*), while cerebellar KL is restricted to Purkinje cells (PCs), which MLIs provide GABAergic synaptic inhibition to. We tested the hypothesis that PC KL and MLI Kit were essential to the GABAergic inhibition of PCs. We created a Kit conditional knockout mouse and accomplished embryonic knockout of Kit from MLIs; separately, we knocked out KL from postnatal PCs. By either method, disruption of the KL-Kit pair produced robust and specific impairments to GABAergic inhibition of PCs. Through sparse postnatal viral manipulation of KL in PCs, we provide evidence that KL functions throughout adulthood to regulate synaptic function. These results demonstrate that cell-type-specific expression of KL and Kit influences functional connectivity in the mammalian brain, informing the long-observed expression of KL-Kit in the brain and suggesting a rationale for the affiliation of Kit loss with neurological impairment.

## Results

We generated mice in which *Kit* exon 4 is flanked by LoxP sites (*Kit* tm1c); *Kit* gene modifications are illustrated in *Figure 1C and D*. Kit tm1c homozygous Control animals had normal appearance, sex ratios, and reproductive success. Kit protein is abundantly expressed in parvalbumin-positive GABAergic interneurons in the molecular layer of the cerebellar cortex (MLIs), whereas KL is expressed by PCs (*Figure 1A and B*). To determine if Kit influences synapse function, we conditionally knocked out Kit from MLIs and assessed the MLI:PC synapse. Pax2 is a transcription factor expressed early in the lineage of GABAergic interneurons of the cerebellum (*Maricich and Herrup, 1999*); a *Pax2*-Cre transgene enables embryonic recombination (*Blake and Ziman, 2014*; *Pfeffer et al., 2002*). We generated *Kit* tm1c homozygous litters of which nominally half were hemizygous for *Pax2* Cre (Kit KO) and half were not (Control). We produced Control and Kit KO animals in normal sex and genotype ratios. As expected for the C57BL/6J background, Control littermates had dark coat and eyes, but littermate Kit KO animals had white whiskers, white fur with variable pigmented patches, and black eyes; example littermates are provided in *Figure 1E*. Given Pax2 expression in melanocytes (*Lee et al., 2011*) and the role of Kit in melanogenesis reviewed in *Wehrle-Haller, 2003*; *Besmer et al., 1993*, the pigmentation phenotype provided gross confirmation of conditional Kit KO. We thus confirmed that cerebella of Kit KO animals were Kit depleted. By immunohistochemistry in Control animals, we recapitulated the known pattern of Kit immunoreactivity in the parvalbumin-positive MLIs of the cerebellar cortex (Control, *Figure 1F* and inset). In contrast, a sex-matched Kit KO littermate demonstrated loss of cerebellar Kit immunoreactivity (Kit KO, *Figure 1F* and inset). In data not shown,

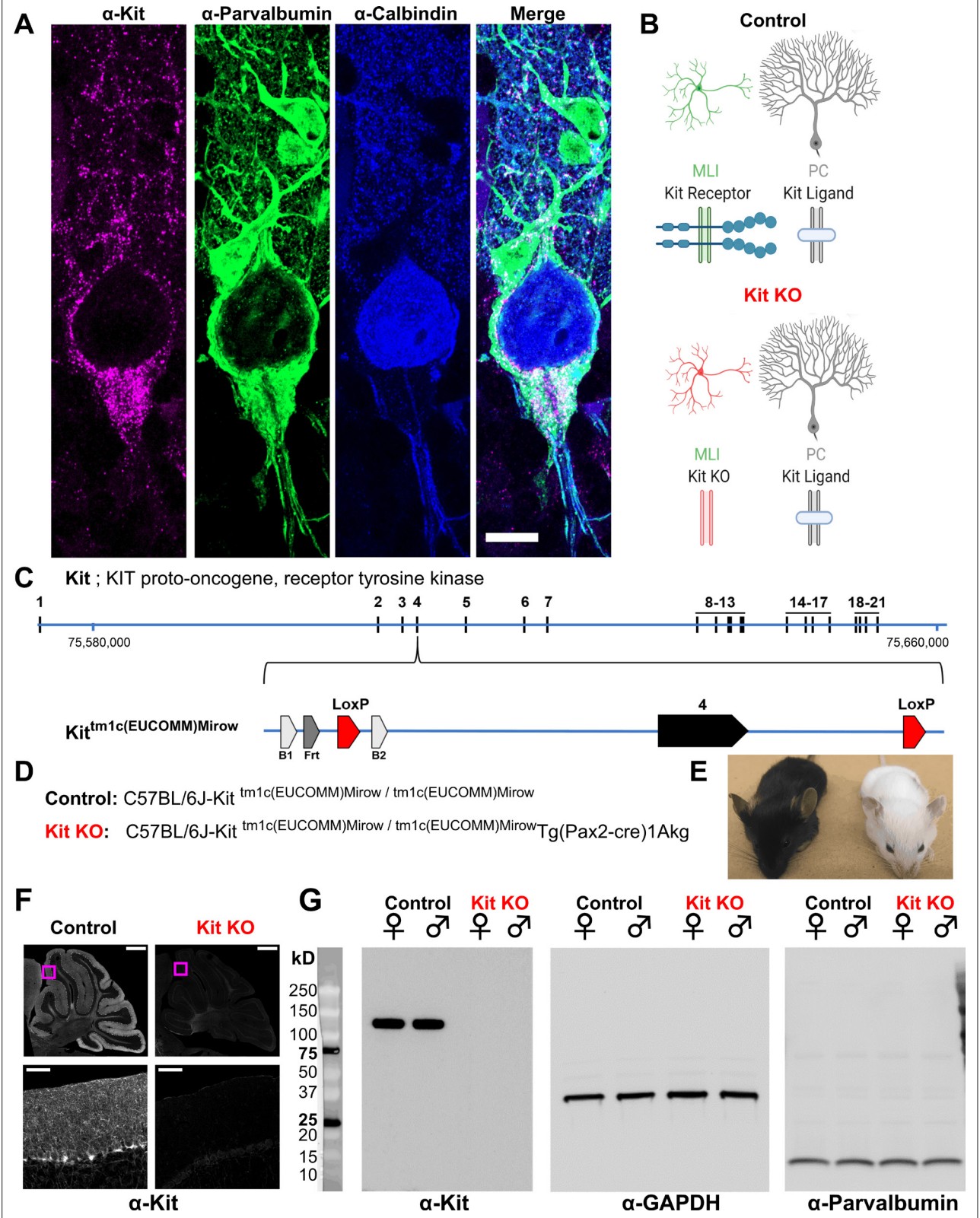

**Figure 1.** Design and validation of a Kit conditional knockout (KO) mouse. (**A, B**) Kit receptor tyrosine kinase (Kit) is enriched in parvalbumin-positive GABAergic interneurons of the molecular layer (i.e., basket and stellate cells, MLIs) of the cerebellar cortex, where they synapse onto each other and onto Purkinje cells (PCs, Calbindin+), which express Kit Ligand (KL). Scale bar 10 microns. Expression pattern schematized in (**B**), Control. (**C**) In humans and mice, Kit protein is encoded by up to 21 exons, which in mouse is encoded on the plus strand of chromosome 5 at 75,735,647–75,817,382 bp. We

*Figure 1 continued on next page*

*Figure 1 continued*

generated a Kit conditional knockout mouse in which *Kit* exon 4 is floxed, flanked by LoxP sites. (**D**) We generated Control mice homozygous for the *Kit* floxed allele *Kit*tm1c(EUCOMM)Mirow, which varied in (*Pax2*-Cre)1Akg transgene status, with the goal of depleting Kit from MLIs in embryonic development. (**E**) *Pax2* Cre-mediated Kit KO mice were notably hypopigmented in hair and whiskers, though not eyes. (**F**) Confocal microscopy of Kit immunoreactivity in cerebella from age and sex-matched Control and Kit KO littermates demonstrates the established enrichment of Kit in the molecular layer of the Control cerebellar cortex, and its loss in Kit KO. Scale Bar 500 microns; top row, 50 microns inset. (**G**) Utilizing a distinct assay and different primary antibody, we confirm the detection of Kit immunoreactivity in Controls, and its loss in Kit KO litter mates of either sex by western blot of total protein lysates of cerebella. We affirmed equivalent protein loading by GAPDH and parvalbumin.

The online version of this article includes the following source data and figure supplement(s) for figure 1:

**Source data 1.** Original unedited unlabeled blots for *Figure 1G*: Kit, GAPDH, and parvalbumin, respectively.

**Source data 2.** Original unedited unlabeled blots for *Figure 1G*: Kit, GAPDH, and parvalbumin, respectively.

**Source data 3.** Original unedited unlabeled blots for *Figure 1G*: Kit, GAPDH, and parvalbumin, respectively.

**Source data 4.** Labeled blots for *Figure 1G*.

**Figure supplement 1.** *Kit* receptor tyrosine kinase is a highly conserved gene affiliated with neurological impairment.

we confirmed that both mGlur1/2 and neurogranin-positive cerebellar Golgi cells (which share the MLI Pax2+ lineage, and of which some are transiently Kit positive) were still present in the Kit KO condition (*Maricich and Herrup, 1999*; *Leto et al., 2006*; *Weisheit et al., 2006*; *Amat et al., 2017*). By western blot of cerebellar lysates from Control animals, Kit resolved as a major band at ~120 kD, but in Kit KO littermates, Kit immunoreactivity was lost; GAPDH as loading control was equivalent, as was parvalbumin (which marks mature MLIs and PCs) (*Figure 1G*).

Having validated that Kit KO animals indeed lacked cerebellar Kit, we then assessed the functional impact. We compared spontaneous GABAergic inhibitory postsynaptic currents (sIPSC) in PCs of acute slices generated from ~P34 Control and Kit KO animals, schema (*Figure 2A*, example traces in *Figure 2B*). Kit KO was associated with a significant change in the distribution of sIPSC event amplitudes (*Figure 2C*). For each PC of Control or Kit KO conditions, we calculated the average sIPSC event frequency and amplitude, as well as total inhibitory charge transfer over the recording epoch. We determined there was a significant decrease in sIPSC frequency (*Figure 2D*) and a less severe decrease of sIPSC amplitude and inhibitory charge. This reduced inhibition did not produce obvious impacts to spontaneous PC firing dynamics: we determined (in 29 Control vs 22 Kit KO cells) that PC membrane potential and membrane resistivity were not significantly different, and that per cell average: spontaneous action potential frequency, inter-spike interval (ISI), and ISI CV2 were not significantly different, despite some difference in the distribution of individual ISI across genotypes. (*Figure 2—figure supplement 1*). Though circuit consequences of the Kit KO-mediated PC disinhibition were thus not immediately apparent, the reduced sIPSC frequency in Kit KO was not associated with changes in the density or distribution of MLIs or in the thickness of the molecular layer (*Figure 2—figure supplement 2*).

Since the number and distribution of MLIs were normal in Kit KO, we sought to determine if the reduced PC inhibition was related to altered MLI physiology (schema, *Figure 2—figure supplement 3A*). However, capacitance, input resistance, and spontaneous or evoked MLI action potential frequency was not different between Control and Kit KO (*Figure 2—figure supplement 3B–E*). Since MLI numbers, distribution, and firing were apparently normal in Kit KO, we sought to determine if there were defects in evoked neurotransmitter release. Direct electrode stimulation of the outer or inner molecular layer in a paired-pulse paradigm evoked inhibitory postsynaptic currents (IPSCs) of equivalent mean amplitude in PCs from Control and Kit KO conditions, and we detected no difference in the paired pulse ratio across genotypes (*Figure 2—figure supplement 3F–M*). These data suggest that electrical stimulation can produce IPSCs of similar amplitude, and that release probability of individual synapses is unchanged, between Control and Kit KO conditions. We therefore investigated markers of the MLI:PC synapses to determine if they were altered.

The density and average size of GABAergic synapses onto PCs (triple-positive VGAT, gephyrin, and calbindin puncta in the molecular layer) was not reduced by Kit KO (example immunofluorescence, *Figure 2—figure supplement 4A*; quantification, *Figure 2—figure supplement 4B and C*). MLI axon terminals contribute not just to individual axo-dendritic synaptic puncta, but also to interdigitated MLI axon collaterals around the soma and initial axon segment of PCs, forming 'pinceaux' structures. By

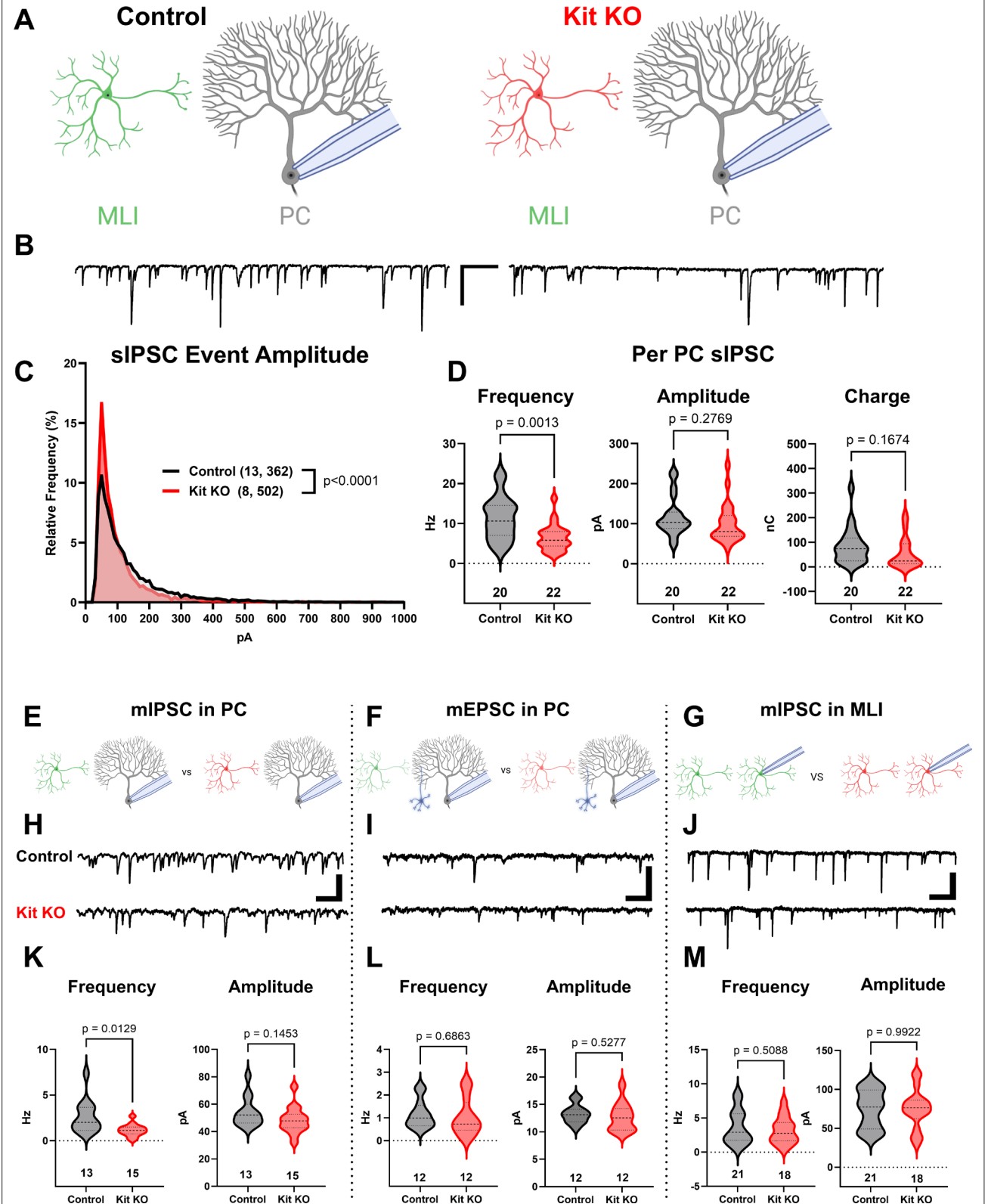

**Figure 2.** The knockout of Kit from cerebellar cortex interneurons impairs inhibition of Purkinje cells (PCs). (**A, B**) Experimental schema and example traces of spontaneous inhibitory postsynaptic currents (sIPSC) in PCs from Control animals or from those with Kit KO from cerebellar cortex molecular layer interneurons (MLIs). Scale bar is 500 ms × 100 pA. (**C**) A frequency distribution plot for individual sIPSC event amplitudes recorded in PCs as in (**A, B**). A Kolmogorov–Smirnov (KS) test reveals a significant difference in the distribution of these event amplitudes; p < 0.0001, n in chart. (**D**) For each PC,

*Figure 2 continued on next page*

*Figure 2 continued*

the average sIPSC frequency and amplitude, and the total inhibitory charge transfer, was determined. There was a significant ~50% decrease in sIPSC frequency, but the decrease in amplitude or charge transfer was not significant. (**E–G**) Experimental schema: in separate experiments, we recorded miniature postsynaptic currents from PCs or from MLIs in Control and Kit KO animals. (**H**) Example traces of miniature inhibitory postsynaptic currents (mIPSCs) or (**I**) miniature excitatory postsynaptic currents (mEPSCs) recorded in PCs, and example traces of mIPSC in MLIs, all from Control or Kit KO animals. Scale for (**H**) and (**J**) is 50 pA, 25 pA for (**I**), all 500 ms. (**K–M**) Analysis of per cell average miniature event frequency and amplitude revealed that Kit KO significantly reduced mIPSC frequency, but not amplitude, in PCs by > 50%, p = 0.013. mEPSCs in PCs and mIPSC in MLIs were not significantly different between Control and Kit KO. n in charts refers to the number of cells. Error bars are SEM. p-Values were calculated by a two-tailed *t*-test, with Welch's correction as needed.

The online version of this article includes the following figure supplement(s) for figure 2:

**Figure supplement 1.** Kit knockout (KO) does not impact basal Purkinje cell (PC) firing.

**Figure supplement 2.** Kit knockout (KO) does not alter molecular layer interneuron (MLI) number or distribution.

**Figure supplement 3.** Kit knockout (KO) does not alter intrinsic properties of molecular layer interneurons (MLIs).

**Figure supplement 4.** The impact of Kit knockout (KO) on the size of molecular layer interneuron (MLI) synaptic structures.

parvalbumin immunoreactivity, these structures were preserved but smaller in Kit KO (arrowheads, *Figure 2—figure supplement 4C*), and this was confirmed by decreased Kv1.2 (*Figure 2—figure supplement 4D*) and PSD-95 (*Figure 2—figure supplement 4E*). As PSD-95 immunoreactivity faithfully follows multiple markers of pinceaux size (*Zhou et al., 2020*), we quantified PSD-95 immunoreactive pinceau area and determined that pinceaux size decreased by ~50% in Kit KO (26 Control vs 43 Kit KO pinceau from n of 5 Control vs 8 Kit KO animals, *Figure 2—figure supplement 4F*). These results suggested that Kit KO MLIs may have defects in physical and/or functional maturation of synapses onto PCs.

To evaluate the number of functional GABAergic synapses, we analyzed miniature synaptic currents in acute cerebellar slices (schema, *Figure 2E–G*; example traces, *Figure 2H–J*). In PCs of Kit KO, there was a significant ~50% reduction in the average frequency, but not amplitude, of miniature inhibitory postsynaptic currents (mIPSCs) (*Figure 2K*). In contrast, the average frequency and amplitude of miniature excitatory postsynaptic current (mEPSC) in PCs from Control and Kit KO did not differ (*Figure 2L*). These data suggested that Kit KO impaired the MLI:PC synapse but did not rule out a general defect in MLI synapse function. As MLIs synapse not just onto PCs, but also onto other MLIs, we evaluated mIPSCs in MLIs. Mean mIPSC frequency and amplitude in MLIs was unchanged between Control and Kit KO (*Figure 2M*). Therefore, embryonic MLI Kit KO was associated with a specific defect in the number or proportion of functional GABAergic synapses onto PCs. As MLI Kit and PC KL expression is maintained postnatally, we hypothesized that postnatal postsynaptic PC KL KO would phenocopy embryonic presynaptic MLI Kit KO.

We utilized previously described *KL* floxed (*Ding et al., 2012*) and *Pcp2*-Cre (*Barski et al., 2000*) strains to yield mice with Control or KL KO PCs (schema, *Figure 3A*). We confirmed depletion of cerebellar KL transcripts by rtPCR in data not shown (*Ding et al., 2012*) and that PC KL KO did not grossly disrupt the pattern of Kit immunoreactivity. Patch clamp of PCs in acute slices revealed that PC KL KO produced significant differences in the distribution of sIPSC event amplitudes (example traces, *Figure 3B and C*). PC KL KO reduced the average sIPSC frequency and amplitude, as well as the total inhibitory charge transfer ('charge') recorded in PCs (*Figure 3D*), without changes to PC capacitance or membrane resistance. The phenotype was specific to GABAergic synapses: there was a substantial reduction in the frequency but not the amplitude of mIPSC events in PCs (*Figure 3E*). As with Kit KO, we detected no decrease in the per-animal average size or density of triple-positive VGAT/gephyrin/calbindin synaptic puncta in the molecular layer (n = 4 Control vs 4 KL KO animals, mean area 0.249 vs 0.286 $\mu m^2$, p = 0.1962; mean density 0.045 vs 0.046 per $\mu m^2$, p = 0.9658, not illustrated). We further detected no difference in PDS-95+ pinceau area (49 Control vs 48 KL KO pinceaux from n = 4 Control vs 4 KL KO animals, mean area 57.93 vs 53.37 $\mu m^2$, p = 0.6036, not illustrated). Thus, as with embryonic Kit KO, postnatal KL KO apparently reduced the proportion of functional, if not the absolute total number, of GABAergic synapse sites upon PCs. In contrast to this impact on GABAergic synapses, the frequency and amplitude of mEPSCs in KL KO PCs were not different vs Controls (*Figure 3F*). In

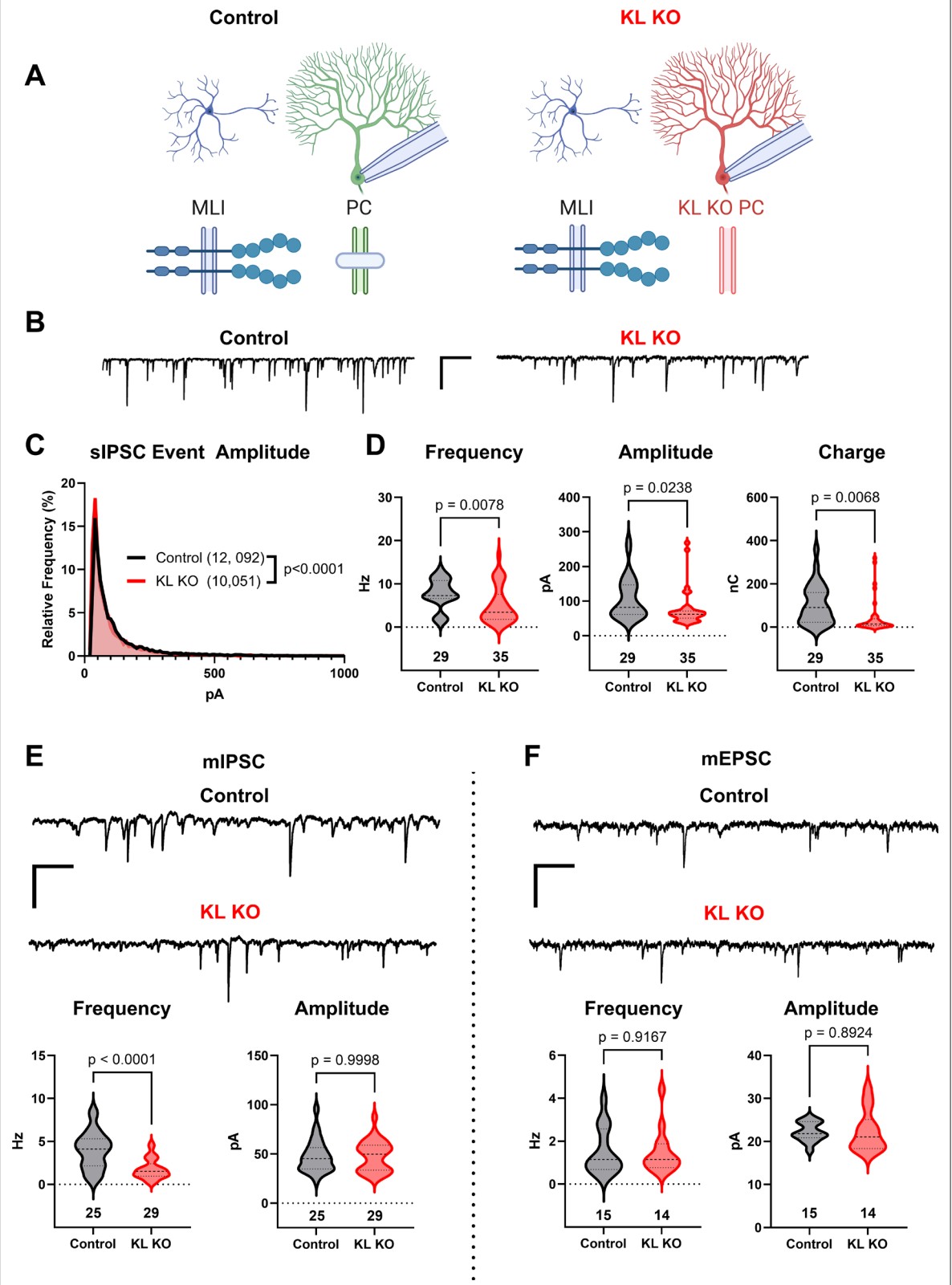

**Figure 3.** Knockout of Kit Ligand (KL KO) from Purkinje cells decreases the inhibitory input they receive. (**A, B**) Experimental schema and example traces of inhibitory postsynaptic currents detected in PCs from Control animals or from those with KL KO accomplished by a *Pcp2*-Cre × KL (*Kitl*) floxed strategy. Scale bar is 500 ms × 100 pA. (**C**) A frequency distribution plot for individual spontaneous inhibitory postsynaptic current (sIPSC) event amplitudes recorded in PCs as in (**A, B**). A KS test reveals a significant difference in the distribution of these event amplitudes; p < 0.0001, n in chart.

*Figure 3 continued on next page*

*Figure 3 continued*

(**D**) For each PC, the average sIPSC frequency and amplitude, and the total inhibitory charge transfer, was determined. There was a significant ~40% decrease in sIPSC frequency, amplitude, and charge transferred to KL KO PCs. (**E, F**) The average miniature inhibitory postsynaptic currents (mIPSC) recorded in PC KL KO vs Control PCs had a > 50% decreased frequency p < 0.0001 without a corresponding decrease in amplitude. The average miniature excitatory postsynaptic current (mEPSC) frequency and amplitude recorded in separate PCs did not differ between Control and KL KO. Scale bar is 500 ms × 100 pA for mIPSC, 500 ms × 50 pA for mEPSC. n in charts refers to the number of cells. Error bars are SEM. p-Values were calculated by a two-tailed *t*-test, with Welch's correction as needed.

data not shown, a KS test revealed a significant difference in the individual mEPSC event amplitude distribution (p = 0.0002, n = 1196 Control vs 1244 KL KO); however, the mean mEPSC amplitude was within 1% between conditions. Similarly, a KS test of mIPSC event amplitudes revealed that the mean mIPSC amplitude was ~10% greater in PC KL KO than in Control (p < 0.0001, n = 5996 Control vs 3958 KL KO). It is therefore unlikely that decreased event amplitude contributed to the specific and robust > 50% decrease in mIPSC frequency detected in PC KL KO. Thus, as with MLI Kit KO, PC KL KO specifically reduced synaptic GABAergic inhibition of PCs.

Recombination by *Pcp2*-Cre begins at ~P7 (*Barski et al., 2000*), whereas *Pax2*-Cre recombination occurs at ~E15 (*Ohyama and Groves, 2004*; *Maricich and Herrup, 1999*). Our results thus suggest that continued expression of the KL-Kit dyad sustains inhibitory drive to PCs. To test this hypothesis (and to avoid transformative effects of Kit overexpression), we tuned KL postnatal expression in vivo.

To determine if acute focal manipulations in KL modulate PC inhibition, we injected replication-defective viruses leveraging the PC-specific L7.6 promoter (*Nitta et al., 2017*) to express Cre in KL floxed animals, and identified Cre-positive KL KO PCs by mCherry expression (schema, *Figure 4A*; example transduction, *Figure 4—figure supplement 1A–C, E–G*). Injections were at P7, P18, or P56; acute slices were generated 2–3 wk after injections. In the P7 neonatal animals, we utilized lentivirus due to its reduced spread vs adeno-associated viral (AAV) particles, which were used for P18 and P56. Patch-clamp recordings in areas of sparse infection revealed that regardless of when KL was depleted, PC KL KO reduced PC sIPSC frequency and inhibitory charge transferred (P7, P18, and P56 in *Figure 4—figure supplement 1D, H, and I*; P18 injections *Figure 4B–D*). We next examined the effects of PC KL overexpression using a complementary strategy. We co-injected L7.6 Cre AAV with AAV encoding EF1α-driven Cre-On mCherry and the membrane-bound isoform of mouse KL (mKL2) via a T2A element to produce PC KL overexpression (OX) (schema, *Figure 4E*). Whether at P18 (shown) or P56 (not shown), KL OX PCs had increased sIPSC frequency, amplitude, and inhibitory charge transferred compared to nearby Control PCs (*Figure 4F–H*). We noted that Control PCs from the PC KL KO paradigm received a similar degree of inhibition (frequency, amplitude, charge) as did PCs from Sham injections. In contrast, while the KL OX PCs received elevated GABAergic input compared to their adjacent Control PCs, the (KL OX) relative local increase in inhibitory drive was not beyond the level of inhibition recorded in PCs from separate Sham animals (*Figure 4F–H*). That is, the local gain in synaptic input to KL OX PCs appears to come at the expense of decreasing inhibition to nearby Control PCs.

## Discussion

For genes essential to survival or reproduction, germline knockout strategies are not feasible for evaluating postnatal physiology. Among such genes, *Kit* is an example of a highly pleiotropic gene, and so the interpretation of global hypomorphs is challenging and the recombination of conditional alleles outside of desired tissues can lead to lethal or sterile phenotypes. By generating a conditional knockout mouse, we were able to produce viable animals with Kit protein depleted from the cerebellum to reveal a role in synapse function.

MLI number, distribution, action potential firing, and the density of GABAergic synaptic puncta onto PCs are all apparently normal in MLI Kit KO. These data suggest that while MLI axon terminals are present in Kit KO, they may be less functional. An increased failure rate between action potential generation and neurotransmitter release would be consistent with the normal firing frequency and synapse puncta density but decreased sIPSC and mIPSC frequency observed. Under the PPR

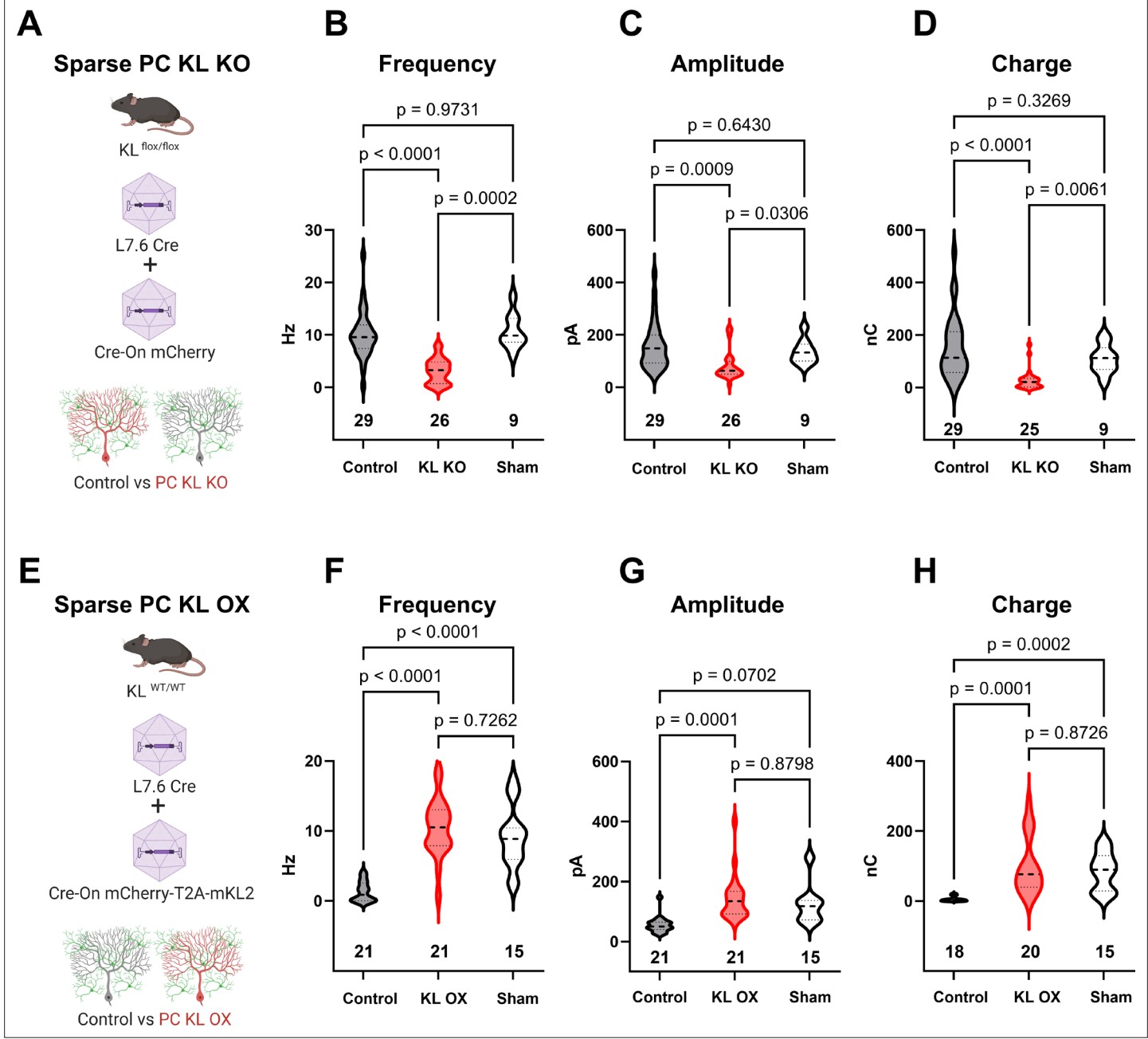

**Figure 4.** Local levels of Kit Ligand influence the inhibition Purkinje cells (PCs) receive. (**A**) To accomplish in vivo Control and sparse Kit Ligand Knockout (KL KO) PCs, an adeno-associated virus (AAV) encoding Cre under the PC-specific L7.6 promoter was co-injected with an AAV encoding a Cre-On mCherry cassette under the Ef1α promoter into mice homozygous for the KL (*Kitl*) floxed allele. (**B–D**) In animals injected at P18, PC KL KO neurons demonstrated an ~70% decrease in spontaneous inhibitory postsynaptic current (sIPSC) event frequency compared to adjacent Control PCs (p < 0.0001) or PCs recorded from Sham control animals (p = 0.0002). Control vs Sham sIPSC frequency was not significantly different. The same pattern was found for both mean sIPSC amplitude (**C**) and for total inhibitory charge transferred (**D**). (**E**) To accomplish in vivo Control and sparse Kit Ligand overexpressing (KL OX) PCs, the L7.6 Cre AAV was co-injected with AAV expressing mCherry and (T2A) murine Kit Ligand isoform 2 under the Ef1α promoter. (**F–H**) In animals injected at P18, PC KL OX neurons demonstrated an approximately eightfold increase in sIPSC frequency vs neighboring Control PCs (p < 0.0001). The frequency of sIPSC events in KL OX PCs was not significantly different from PCs recorded in different Sham animals; this Sham PC sIPSC frequency (while comparable across studies) was sevenfold higher than Control PCs within the KL OX experimental animals (p < 0.0001). A similar pattern was found for sIPSC amplitude (**G**) and total inhibitory charge transferred (**H**). n in charts refers to the number of cells. Error bars are SEM. p-Values were calculated by Brown–Forsythe ANOVA with Dunnett's T3 multiple-comparisons test.

*Figure 4 continued on next page*

*Figure 4 continued*

The online version of this article includes the following figure supplement(s) for figure 4:

**Figure supplement 1.** Sparse acute depletion of Kit Ligand reduces GABAergic input to Purkinje cells (PCs).

paradigm, the apparently equivalent evoked responses may be due to direct stimulation of axon terminals and/or simultaneous recruitment of multiple MLIs.

Kit KO most directly impacts the MLI:PC synapse, based on seemingly normal miniature events for parallel fiber: PC synapses (mEPSC in PC) and MLI:MLI (mIPSC in MLI) synapses. This is perhaps expected given the cell-type-specific expression of KL by PCs and Kit by MLIs; however, it is a critical distinction as cerebellins (*Zhang et al., 2015*) and calsystenin (*Liu et al., 2022*) impact multiple forms of synaptic input to PCs. It will be informative to determine if the residual inhibitory drive to PCs observed in MLI Kit KO or PC KL KO comes from PC axon collaterals *Chan-Palay, 1971*; *Larramendi and Lemkey-Johnston, 1970*; *Orduz and Llano, 2007*; *Witter et al., 2016* whose input we might expect to be preserved if KL-Kit functions primarily to mediate connectivity between different cell types or perhaps from a subtype of interneuron not dependent upon Kit signaling. This distinction is further relevant since our histological analysis of synaptic puncta does not resolve between MLI:PC and PC:PC synapses, the relative balance of which may be altered by either KL or Kit manipulations. Indeed, future studies of the role of KL-Kit on PC-PC connectivity may inform how PC KL OX can apparently weaken inhibition to adjacent PCs: this may occur through the reduction of inhibitory drive from KL OX PCs to adjacent control PCs. An alternative and not mutually exclusive possibility is that PC KL OX excessively potentiates MLI Kit to enhance MLI:MLI inhibition, reducing the inhibitory output surrounding MLIs can provide to control PCs.

Though such important nuances of the circuit mechanisms remain to be elucidated, our results demonstrate that the KL-Kit axis, from embryonic development through young adulthood, can alter GABAergic drive to PCs. These results suggest that KL-Kit may be necessary to maintain, and perhaps capable of modulating, MLI:PC GABAergic inhibition in the adult cerebellum. As each MLI (e.g., basket cell) has axon collaterals that can target multiple PCs, and as each PC can receive input from multiple MLIs, it is interesting to speculate that dynamics (e.g., in the expression or shedding) of KL may act through presynaptic Kit to tune the degree of convergent MLI:PC inhibitory drive. This would complement existing forms of modulation at the MLI:PC synapse, such as those dependent on other PC-derived signaling molecules, like endocannabinoids (*Kreitzer et al., 2002*; *Kreitzer and Regehr, 2001*; *Safo et al., 2006*) or peptides like secretin (*Williams et al., 2012*; *Fuchs et al., 2014*) and CR (*Wu et al., 2020*). The physiology of PCs, and the structure of MLI:PC inhibition, also varies over zones such as those identified by Zebrin-II and PLC β4 (*Zhou et al., 2014*; *Brochu et al., 1990*; *Leclerc et al., 1990*; *Sarna et al., 2006*; *Marzban et al., 2007*); whether there is functional variation in KL or Kit expression, signaling, or dependency remains to be determined. Though we have demonstrated a role in synaptic GABAergic inhibition, it is currently unknown whether Kit signaling also plays a role in other forms of inhibition in the cerebellar cortex, such as MLI:MLI gap-junctions, or MLI:PC ephaptic coupling. The latter may be suggested by the Kit KO-mediated reduction in Kv1.1/Kv1.2 and PSD-95 immunoreactivity.

While our data clearly demonstrates that KL-Kit influences synaptic function, we do not yet know the molecular mechanisms involved. Kit is a receptor tyrosine kinase that can activate multiple downstream effectors, including Src family kinases, PLC, PI3K/mTOR, and the MAPK pathway, the latter two notable here for their shared affiliation with synapse phenotypes and autism spectrum disorder. Investigation of the interactome of Kit under basal and KL manipulated conditions may inform the receptor kinase cascades through which Kit signaling sustains synaptic function. It also plausible that trans-cellular interactions between KL and Kit may function in a kinase-independent, cell adhesion molecule modality (*Adachi et al., 1992*; *Tabone-Eglinger et al., 2014*). There is a substantial literature illustrating the importance of adhesion molecules in the organization and maintenance of MLI:PC inhibition, such as neurexins/neuroligins, dystrophin/dystroglycan, semaphorins, and neurofascins (*Zhang et al., 2015*; *Liu et al., 2022*; *Ango et al., 2004*; *Buttermore et al., 2012*; *Telley et al., 2016*; *Cioni et al., 2013*; *Wu et al., 2022*). Much of what is understood about these molecules in the maintenance of PC inhibition is revealed by postsynaptic phenotypes; that is, *cis-* and *trans* interacting proteins that, via gephyrin, maintain the organization of PC GABA$_A$ receptors. It is notable therefore

that MLI Kit or PC KL KO has a more predominant effect on mIPSC frequency than on amplitude, suggesting a presynaptic phenotype.

Despite the reduced PC inhibition resulting from MLI Kit KO or from PC KL KO, we have not observed overt cerebellar signs (i.e., ataxia). We suspect that this is due to compensation. Even the developmental ablation of functional GABA$_A$ channels in PCs does not produce overt motor signs, while adulthood ablation can (*Wisden et al., 2009*). Thus, we might unmask behavioral phenotypes if KL or Kit KO is delayed until adulthood. Consistent with the interpretation that developmental compensation may be involved, the impact of postnatal PC KL KO on inhibitory drive to PCs is more pronounced than embryonic MLI Kit KO. Beyond direct inhibition of PCs, MLI:PC GABAergic inhibition constrains granule cell-mediated PC dendritic excitation and thus gates the conversion of PC LTD to LTP at the PF-PC synapse in motor learning (*Bonnan et al., 2023*; *Gaffield et al., 2018*). It is therefore possible that KL or Kit KO may only reveal phenotypes in motor learning (or other cerebellar-plasticity dependent) behaviors; speaking to the feasibility of this possibility, global hypomorphs for KL or Kit have altered hippocampal neurophysiology and learning performance on a Water Maze task (*Motro et al., 1996*; *Katafuchi et al., 2000*; *Kondo et al., 2002*). Continuous expression of KL and Kit in specific neurons may thus reflect not only a role in the development (or maintenance) of synaptic function as suggested here but also in the plastic modulation of synaptic strength.

Despite the importance of future works to determine molecular mechanisms and organismal consequences, our study is a discrete advance because it establishes that KL-Kit regulates mammalian central synapse function. Here, we demonstrate impacts at the mouse GABAergic MLI:PC synapse; however, Kit and KL are expressed in diverse organisms and neuronal populations including glutamatergic neurons of the hippocampus, and neurons in olfactory bulb, basal forebrain, and brainstem. It is therefore feasible that the synaptic phenotypes reported here reflect that KL-Kit is broadly capable of influencing connectivity.

## Materials and methods
### Animals
All procedures performed at Michigan State University were approved by the Institutional Animal Care and Use Committees of Michigan State University, which is accredited by the Association for Assessment and Accreditation of Laboratory Animal Care. Mice were on a 12 hr dark/light cycle, with ad libitum food and water. Male and female mice were used for all experiments. Mice were genotyped in-house using standard PCR-based methods, or via Transnetyx.

### Creation of Kit conditional knockout mouse
We generated a strain of mice having exon 4 of *Kit* flanked by LoxP sites ('floxed'). This was accomplished by utilizing the services of UNC Animal Models Core (Dr. Dale Cowley) to generate chimeras from an ES cell line having a knockout-first allele (with conditional potential) for *Kit* (HEPD0509_8). The ES clones were obtained through the European Mouse Mutant Cell Repository of the European Conditional Mouse Mutagenesis Program (EUCOMM) (*Skarnes et al., 2011*). Chimeric mice were screened for germline transmission of the targeted allele and were then crossed to an Flp deleter strain (C57BL/6N-Albino-Rosa26-FlpO) to convert the knockout-first allele to a conditional allele (tm1c) through removal of the FRT-flanked selection cassette. F1 male offspring carrying *Kit* tm1c and Flp alleles were crossed to C57BL/6J females; we selected resultant progeny heterozygous for *Kit* tm1c but negative for Flp and the selection cassette. The *Kit* tm1c allele was bred to homozygosity while selecting against animals carrying the Tyr c-Brd allele (previously conveyed by C57BL/6N-Albino-Rosa26-FlpO). The resultant strain of mice having the *Kit* floxed tm1c allele is given the nomenclature C57BL/6N−*Kit*[tm1c(EUCOMM)Mirow]/J; the Mirow lab code is registered through the National Academies to the author and principal investigator herein, **Mi**chael **Ro**land **W**illiams. Additional strains of mice used were *Pax2*-Cre transgenic mice STOCK Tg(*Pax2*-Cre)1Akg/Mmnc, RRID:MMRRC_010569-UNC *Ohyama and Groves, 2004*; *Kit* eGFP (Tg(*Kit*-EGFP)IF44Gsat/Mmucd; Gensat) [(*Heintz, 2004*)]) backcrossed for greater than five generations to C57BL/6J (stock# 000664, The Jackson Laboratory), *Kitl* (KL) floxed (*Kitl* tm2.1Sjm/J, stock# 017861) (*Ding et al., 2012*), and *Pcp2*-Cre (stock# 004146) mice (*Barski et al., 2000*), via The Jackson Laboratory.

To produce mixed litters of 'Control' and 'Kit KO' animals, mice homozygous for the *Kit* tm1c allele but negative for *Pax2*-Cre were crossed to mice homozygous for the floxed allele and *Pax2*-Cre positive. We used Kit KO females crossed to Control males as our breeding scheme. Experimental animals were derived from at least five generations of backcrossing Kit KO animals to *Kit* tm1c homozygous animals derived from a separate (*Kit* tm1c homozygous × *Kit* tm1c homozygous) breeding cohort.

## Validation of Kit knockout

We investigated conditional knockout of Kit protein from the cerebellum by western blot of total protein lysates from acutely harvested cerebellums (~P52) and indirect immunofluorescence on free-floating vibratome-generated sections of transcardial-perfused formaldehyde-fixed mouse brains (~P31). Western blots were probed with rabbit: α-Kit (1:1000, clone D13A2, product 3074 from Cell Signaling Technologies, RRID:AB_1147633), α-GAPDH (1:2000, clone 14C10, product 2118 from Cell Signaling Technology, RRID:AB_561053), or α-parvalbumin antibody (1:3000, product PV-27 from Swant, RRID:AB_2631173), each primary was detected via HRP-conjugated goat α-rabbit secondary antibody (1:4000, Bio-Rad product 170-6515, RRID:AB_11125142) after incubation with Clarity ECL substrate (Bio-Rad product 1705061), via a Bio-Rad ChemiDoc MP.

For subsequent imaging and electrophysiology, we focused on lobule IV/V of the cerebellar cortex as this area was superficially accessible for surgery and provided a consistent area for focused comparisons between animals that would reduce the influence of spatial variations in KL or Kit expression.

For indirect immunofluorescence, primary antibodies were rat α-Kit (1:500, MA5-17836 clone ACK4 from Thermo Fisher, RRID:AB_2539220), mouse α-calbindin (1:2000, α-calbindin-D28K, C9848 clone CB-955 from Sigma-Aldrich, RRID:AB_476894), and rabbit α-parvalbumin (1:500, PV-27 from Swant, RRID:AB_2631173). Primaries were detected by goat-host highly cross-adsorbed secondary antibodies coupled to Alexa 488, Cy3, or Alexa 647 (1:400, Jackson ImmunoResearch). Nuclei were labeled with DAPI, tissues were mounted in an anti-fade reagent, and fluorescent signals captured laser scanning confocal microscopy. Cell densities for MLIs (parvalbumin-positive and/or calbindin-negative somas of the molecular layer) were performed on maximum Z-projections of 2.5–2.6-micron-thick stacks using the multipoint feature for counting somas, and the polygon ROI tool for determining the area of the Molecular Layer in which the somas were counted; results from two technical replicates were averaged for each animal. The polygon tool was also used for determining maximal PSD-95+ pinceau area. To determine the width of the molecular layer, seven linear ROIs per field of view of comparable locations of lobules IV/V were averaged to generate per-animal layer thickness measurements. For analysis of synapses, we quantified puncta (of 0.05–5 $\mu m^2$ using the Analyze Particles function) in the molecular layer that were triple positive (each channel autothreshold) for rabbit α-VGAT(1:500, 131003, RRID:AB_887869), mouse α-Gephyrin (1:5000, 147011, RRID:AB_887717), and chicken α-calbindin (1:500, 214006, RRID:AB_2619903), all from Synaptic Systems. For analysis of pinceau markers, we utilized mouse α-Kv1.1 (1:250, K36/15, RRID:AB_2877294), Kv1.2 (1:250, K14/16, RRID:AB_2877295), and PSD-95 (1:200, K28/43, RRID:AB_2877189), all from NeuroMab/Antibodies Inc.

## Stereotaxic procedures

Injections of replication-defective viral particles used protocols substantially similar to those previously published (*Fricano-Kugler et al., 2016*). In brief, mice were brought to an anesthetic plane using inhaled isoflurane by a low-flow system (SomnoSuite, Kent); thermal regulation was assisted by heating pad, and pain managed through topical lidocaine and by intraperitoneal ketoprofen. A digital stereotaxic device (Kopf) and reference atlas were used to localize a small hole in the skull; cerebellar coordinates were neonates; A/P, –2.55; M/L, –1.1 (reference Lambda), D/V, 1.5–0.5 from dura; juvenile; A/P, –6.25; M/L, –1.35; D/V, 1.5–0.75 (reference Bregma); adults; A/P, –6.35; M/L, –1.8; D/V, 1.5–0.75 (reference Bregma). Viral particles were delivered via an ~30-gauge Hamilton Syringe controlled by digital syringe pump (WPI). The infused virus volume was 1–2 ul, and there was a 5 min period after infusion prior to needle withdrawal. Animals were monitored during and after operations. Ages for injection were, nominally, postnatal day (P) 7, 19, and 56; for P7 and P19, variability of up to 2 d was utilized to accommodate differences in animal size, and for P56 the eighth week of life was utilized.

## Viruses

Lentiviral particles were derived from our vectors previously described to express transgenes under the hUbiC promoter and pseudotyped with VSV-G (*Luikart et al., 2011*; *Williams et al., 2015*). Silent mutations disrupted the internal EcoRI sites of mouse *Kitl* (MC204279, OriGene), the *Kitl* coding sequence was then flanked by EcoRI sites, and the subsequent *Kitl* coding sequence was introduced into a FUC-T2A-(EcoRI) plasmid, to create a FUC-T2A-KL-1 plasmid, wherein C designates mCherry. Subsequent deletion of *Kitl* Exon 6 generated a KL-2 vector (*Martin et al., 1990*; *Huang et al., 1992*; *Flanagan et al., 1991*; *Langley et al., 1994*). Sequences were verified by sequencing. AAV particles were created by Vigene to drive Cre expression by the PC-specific L7.6 promoter (*Nitta et al., 2017*) or drive mCherry or mCherry-T2A-KL-2 in a Cre-On fashion under the EF1α promoter, with the AAV1 serotype.

## Electrophysiology

For patch-clamp electrophysiology of Control vs Kit KO, cells were recorded in slices generated from animals at 36.9 ± 0.6 and 37.4 ± 0.9 d for MLIs and PCs, respectively. Analysis of the *Pcp2* Cre-mediated KL KO tissues was conducted similarly at ~40–44 days old. For viral PC KL KO animals injected as neonates (P7), juvenile (P19), and adults (P56) were recorded at an average of ~35, 45, and 83 days old, respectively. Avertin-anesthetized mice were perfused with a carbogen-equilibrated ice-cold slicing solution containing (in mM) 110 $C_5H_{14}ClNO$, 7 $MgCl_2.6H_2O$, 2.5 KCl, 1.25 $NaH_2PO_4$, 25 $NaHCO_3$, 0.5 $CaCl_2$-$2H_2O$, 10 glucose, and 1.3 Na-ascorbate. In the same solution, we generated 250-micron-thick parasagittal sections of the cerebellum (Leica VT1200). Slices recovered at 34°C for 30 min in a carbogenated ACSF containing (in mM) 125 NaCl, 25 $NaHCO_3$, 1.25 $NaH_2PO_4$, 2.5 KCl, 1 $MgCl_2.6H_2O$, 1 $CaCl_2$-$2H_2O$, and 25 glucose; slices were then held at room temperature for at least 30 min before recording. Recordings were performed in carbogenated external recording ACSF (32.7 ± 0.1°C) containing (in mM) 125 NaCl, 25 $NaHCO_3$, 1.25 $NaH_2PO_4$, 2.5 KCl, 1 $MgCl_2.6H_2O$, 2 $CaCl_2$-$2H_2O$, and 25 glucose. MLIs and PCs of folia IV/V were targeted for recording by IR-DIC, or epifluorescence stimulated by a CoolLED pE-4000, detected via a SciCam Pro on a SliceScope Pro 6000-based rig (Scientifica). Recording electrodes were pulled (Narishige, PC-100) from standard-wall borosilicate glass capillary tubing (G150F-4, Warner Instruments) and had 5.1 ± 0.08 and 2.8 ± 0.02 MΩ tip resistance for MLIs and PCs, respectively. sIPSCs and mIPSC were recorded with an intracellular solution containing (in mM)140 CsCl, 4 NaCl, 0.5 $CaCl_2$-$2H_2O$, 10 HEPES, 5 EGTA, 2 Mg-ATP, and 0.4 Na-GTP, 2 QX-314. For action potential recordings and mEPSCs, the intracellular solution contained (in mM) 140 K-gluconate, 10 KCl, 1 $MgCl_2$, 10 HEPES, 0.02 EGTA, 3 Mg-ATP, and 0.5 Na-GTP as described previously (*Zaman et al., 2011*). The internal pipette solution pH was adjusted to 7.35 with CsOH for IPSCs and mIPSCs, and with KOH for mEPSCs and action potentials, while for all the osmolarity was adjusted to 300 mOsmolL-1 with sucrose. In whole-cell voltage-clamp mode, PCs were held at –70 mV; to isolate sIPSCs, CNQX or NBQX (10 µM) and D-AP5 (50 µM) were added to the recording solution and mIPSC recordings additionally included 1 µM tetrodotoxin (TTX). To record mEPSCs, 1 µM TTX and 50 µM picrotoxin were added to extracellular solution. Action potentials were recorded in MLIs within the middle third of the molecular layer; spontaneous action potentials were recorded in I = 0 mode, or action potentials were evoked by depolarizing current injection in 600 ms steps of 10 pA, from 0 to 150 pA, with a 7 s inter-sweep interval. For paired pulse, current ranging from 150 uA to 250 uA was delivered by concentric bipolar stimulating electrode in the molecular layer. Stimulus duration was 50 µS and ISI was 50 ms. For each cell, 10 sweeps were recorded with 10 S inter-sweep interval. A square-wave voltage stimulation pulse was utilized to determine input resistance and cell capacitance. PCs with an access resistance of 10–20 (15.9 ± 0.5) or MLIs with an access resistance < 30 (23.6 ± 0.6) MΩ were considered for recording. Recordings with > 20% change in series resistance were excluded from analysis. Signals were acquired at 10 kHz with a low-noise data acquisition system (Digidata 1550B) and a Multiclamp700-A amplifier and were analyzed using pClamp11.1 (Molecular Devices) after low pass Bessel (eight-pole) filtration (3 dB cutoff, filter 1 kHz). The minimum amplitude threshold for detecting IPSCs and EPSCs was 15 pA; for mIPSCs and mEPSCs, the cutoff was 8 pA. For determining frequency and amplitude, individual events longer than 1 ms were included while overlapping events were manually rejected from analysis. Inhibitory charge is reported as the per-cell sum over the recording epoch of the area under the curve for all pharmacologically resolved inhibitory events. For the analysis of minis, both mIPSCs and mEPSCs with over 8 pA amplitude and longer than

1 ms duration were included while overlapping events were rejected from analysis; also rejected from analysis was any miniature event for which amplitude, rise, and decay metrics were not all available.

### Software

Microscopic image processing was conducted via FIJI/ImageJ, electrophysiological data was analyzed by pClamp11.1, statistical analyses were conducted via GraphPad Prism 9, and figures were created with BioRender, PowerPoint, and GraphPad Prism 9.

### Data and materials availability statement

Vector plasmids, replication -defective viral particles, and the C57BL/6N−$Kit^{tm1c(EUCOMM)Mirow}$/J mouse strain are all available to qualified non-profit investigators by contacting the corresponding author, Dr. Michael R. Williams. Vector maps / sequences are available at FigShare, https://figshare.com/account/articles/25438570; DOI: 10.6084/m9.figshare.25438570 .

## Acknowledgements

This work was supported by NIH/NIMH K99/R00 MH110665; The Mall Family Foundation; MSU Global Impact Initiative. The research reported in this publication was supported by the National Institute of Mental Health of the National Institutes of Health under Award Number R00MH110665. The content is solely the responsibility of the authors and does not necessarily represent the official views of the National Institutes of Health.

## Additional information

### Funding

| Funder | Grant reference number | Author |
| --- | --- | --- |
| National Institute of Mental Health | R00MH110665 | Michael R Williams |
| The Mall Family Foundation | | Michael R Williams |

The funders had no role in study design, data collection and interpretation, or the decision to submit the work for publication.

### Author contributions

Tariq Zaman, Data curation, Investigation, Methodology, Writing – review and editing; Daniel Vogt, Investigation, Writing – review and editing; Jeremy Prokop, Data curation, Formal analysis, Investigation, Visualization, Methodology; Qusai Abdulkhaliq Alsabia, Gabriel Simms, Investigation; April Stafford, Formal analysis, Investigation, Methodology; Bryan W Luikart, Conceptualization, Supervision, Methodology, Writing – review and editing; Michael R Williams, Conceptualization, Formal analysis, Investigation, Methodology, Supervision, Visualization, Writing – review and editing

### Author ORCIDs

Tariq Zaman ⓘ https://orcid.org/0000-0003-1025-8542
Jeremy Prokop ⓘ http://orcid.org/0000-0002-1818-9695
Bryan W Luikart ⓘ https://orcid.org/0000-0002-3181-6075
Michael R Williams ⓘ http://orcid.org/0000-0003-3310-7181

### Ethics

This study was conducted in accordance with the Guide for the Care and Use of Laboratory Animals of the National Institutes of Health. Animal studies were conducted according to the approved institutional animal care and use committee (IACUC) protocols 202200116 and 201800201 of Michigan State University, which holds Association for the Assessment and Accreditation of Laboratory Animal Care (AAALAC) accreditation.

Reviewer #1 (Public Review): https://doi.org/10.7554/eLife.89792.3.sa1
Reviewer #2 (Public Review): https://doi.org/10.7554/eLife.89792.3.sa2
Reviewer #3 (Public Review): https://doi.org/10.7554/eLife.89792.3.sa3
Author Response https://doi.org/10.7554/eLife.89792.3.sa4

## Additional files

### Supplementary files
• MDAR checklist

### Data availability
In this manuscript, new analysis was performed on previously generated publically accessible dataset(s). The data compilation and analysis was performed by Dr. Jeremy Prokop. The datasets used are detailed at https://doi.org/10.6084/m9.figshare.25000106.v1.

The following dataset was generated:

| Author(s) | Year | Dataset title | Dataset URL | Database and Identifier |
|---|---|---|---|---|
| Prokop JW | 2024 | KIT genomic analysis | https://doi.org/10.6084/m9.figshare.25000106.v1 | figshare, 10.6084/m9.figshare.25000106.v1 |

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
