## [Editor Report · eLife assessment]

This **valuable** study from Zaman et al. demonstrates that the cKit-Kit ligand complex is necessary for the formation and/or maintenance of molecular layer interneuron synapses in cerebellar Purkinje cells. The evidence presented is **convincing**; in particular, the use of cell-type-specific knockout of cKit in molecular layer interneurons and knockout of Kit ligand in Purkinje cells provides robust evidence. This work will be of particular relevance to those interested in inhibitory synapse formation or the role of inhibition in Purkinje cell behavior.

---

## [Referee Report · Reviewer #1 (Public Review)]

This manuscript from Zaman et al., investigates the role of cKit and Kit ligand in inhibitory synapse function at molecular layer interneuron (MLI) synapses onto cerebellar Purkinje cells (PC). cKit is a receptor tyrosine kinase expressed in multiple tissues, including select populations of neurons in the CNS. cKIt is activated by Kit ligand, a transmembrane protein typically expressed at the membrane of connected cells. A strength of this paper is the use of cell-specific knockouts of cKit and Kit ligand, in MLIs and PCs, respectively. In both cases, the frequency of spontaneous or miniature (in the presence of TTX) IPSCs was reduced. This suggests either a reduction in the number of functional inhibitory release sites or reduced release probability. IPSCs evoked by electrical stimulation in the molecular layer showed no change in paired-pulse ratio, indicating release probability is not changed in the cKit KO, and favoring a reduction in the number of release sites. Changes in IPSC amplitude were more subtle, with some analyses showing a decrease and others not. These data suggest that disruption of the cKit-Kit ligand complex reduces the number of functional synapses with only minor changes in synapse strength.

---

## [Referee Report · Reviewer #2 (Public Review)]

In their study, Zaman et al. demonstrate that deletion of either the receptor tyrosine kinase Kit from cerebellar interneurons or the kit ligand (KL) from Purkinje cells reduces the inhibition of Purkinje cells. They delete Kit or KL at different developmental time points, illustrating that Kit-KL interactions are not only required for developmental synapse formation but also for synapse maintenance in adult animals. The study is interesting as it highlights a molecular mechanism for the formation of inhibitory synapses onto Purkinje cells.

The tools generated, such as the floxed Kit mouse line and the virus for Kit overexpression, may have broader applications in neuroscience and beyond.

One general weakness is that Kit expression is not limited to molecular layer interneurons but also extends to the Purkinje layer and Golgi interneurons. But this expression does not conflict with the principal conclusions, as Purkinje layer interneurons form few or no synapses onto Purkinje cells.

In summary, the data support the hypothesis that the interaction between Kit and KL between cerebellar Molecular Layer Interneurons and Purkinje Cells plays a crucial role in promoting the formation and maintenance of inhibitory synapses onto PCs. This study provides valuable insights that could inform future investigations on how this mechanism contributes to the dynamic regulation of Purkinje cell inhibition across development and its impact on mouse behavior.

---

## [Referee Report · Reviewer #3 (Public Review)]

Summary: Bidirectional transsynaptic signaling via cell adhesion molecules and cell surface receptors contributes to the remarkable specificity of synaptic connectivity in the brain. Zaman et al., investigates how the receptor tyrosine kinase Kit and its trans-cellular kit ligand regulate molecular layer interneuron (MLI)- Purkinje cell (PC) connectivity in the cerebellum. Presynaptic Kit is specific for MLIs, and forms a trans-synaptic complex with Kit ligand in postsynaptic PC cells. The authors begin by generating Kit cKOs via an EUCOMM allele to enable cell-type specific Kit deletion. They cross this Kit cKO to the MLI-specific driver Pax2-Cre and conduct validation via Kit IHC and immunoblotting. Using this system to examine the functional consequences of presynaptic MLI Kit deletion onto postsynaptic PC cells, they record spontaneous and miniature synaptic currents from PC cells and find a selective reduction in IPSC frequency. Deletion of Kit ligand from postsynaptic PC cells also results in reduced IPSC frequency, together supporting that this trans-synaptic complex regulates GABAergic synaptic formation or maturation. The authors then show that sparse Kit ligand overexpression in PCs decreases neighboring uninfected control sIPSCs in a potential competitive manner.

Strengths: Overall, the study addresses an important open question, the data largely supports the authors conclusions, the experiments appear well-performed, and the manuscript is well-written. I just have a few suggestions to help shore up the author's interpretations and improve the study.

Weaknesses:

The strong decrease in sIPSC frequency and amplitude in control uninfected cells in Figure 4 is surprising and puzzling. The competition model proposed is one possibility, and I think the authors need to do additional experiments to help support or refute this model. The authors can conduct similar synaptic staining experiments as in Fig S4 but in their sparse infection paradigm, comparing synapses on infected and uninfected cells. Additional electrophysiological parameters in the sparse injection paradigm, such as mIPSCs or evoked IPSCs, would also help support their conclusions.

The authors should validate KL overexpression and increased cell surface levels using their virus to support their overexpression conclusions.

---

## [Author Response]

The following is the authors’ response to the original reviews.

**Reviewer #1 (Recommendations for The Authors):**
Major comments:1. The immunolabeling data in Figure S4 shows no change in puncta number but reduced puncta size in Kit KO. sIPSC data show reduced frequency but little change in amplitude. These data would seem contradictory in that one suggests reduced synaptic strength, but not number, and the other suggests reduced synapse number, but not strength. How do the authors reconcile these results?

Regarding the synaptic puncta, In Kit KO (or KL KO), we have not detected an overt reduction in the average VGAT/Gephyrin/Calbindin positive puncta density or puncta size per animal. With respect to puncta size, only in the Kit KO condition, and only when individual puncta are assessed does this modest (~10%) difference in size become statistically significant. In the revision, we eliminate this figure and focus on the per animal averages.

We interpret that the reduction in sIPSC and mIPSC frequency likely stems from a decreased proportion of functional synapse sites. The number of MLIs, their action potential generation, the density of synaptic puncta, and the ability of direct stimulation to evoke release and equivalent postsynaptic currents, are all similar in Control vs Kit KO. It is therefore feasible that a reduced frequency of postsynaptic inhibitory events is due to a reduced ability of MLI action potentials to invade the axon terminal, and/or an impaired ability for depolarization to drive (e.g. coordinated calcium flux) transmitter release. That is, while the number of MLIs and their synapses appear similar, the reduced mIPSC frequency suggests that there is a reduced proportion of, or probability that, Kit KO synapse sites that function properly.

1. Related to point 1, it would be helpful to see immunolabeling data from Kit ligand KO mice? Do these show the same pattern of reduced puncta size but no change in number?

Although we have not added a figure, we have now added experiments and a corresponding analysis in the manuscript. As we had previously for Kit KO, we now for KL KO conducted IHC for VGAT, Gephyrin, and Calbindin, and we analyzed triple-positive synaptic puncta in the molecular layer of Pcp2 Cre KL KO mice and Control (Pcp2 Cre negative, KL floxed homozygous) mice. We did not find a gross reduction in the average synaptic puncta size or density, or in the PSD-95 pinceau size. From this initial analysis, it appears that the presynaptic hypotrophy is more notable in the receptor than in the ligand knockout. We speculate that this is perhaps because the Kit receptor may have basal activity in the absence of Kit ligand, that Kit may serve a presynaptic scaffolding role that is lost in the receptor (but not the ligand) knockout, or simply that the embryonic timing of the Pax2 Cre vs Pcp2 Cre recombination events is more relevant to pinceaux development, especially as basket cells are born primarily prenatally.

1. The data using KL overexpression in PC (figure 4E,F) are intriguing, but puzzling. The reduction in sIPSC frequency and amplitude in the control PC is much greater than seen in the Kit or KL KO. The interpretation of these data, "Thus, KL-Kit levels may not set the number of MLI:PC release sites, but may instead influence the proportion of synapses that are functional for neurotransmission (Figure 4G)" is not clear and the reasoning here should be explained in more detail, perhaps in the discussion.

We have attempted to clarify this portion of the manuscript by eliminating the cartoon of the proposed model, and by revising and adding to the discussion. Either MLI Kit KO or PC KL KO seems to preserve the absolute number of MLI:PC anatomical synapse sites (IHC) but to reduce the proportion of those synapse that are contributing to neurotransmission (mIPSC). We speculate that sparse PC KL overexpression (OX) may either (1) weaken inhibition to surrounding control PCs by either diminishing KL OX PC to KL Control PC inhibition, and/or (2) act retrogradely through MLI Kit to potentiate MLI:MLI inhibition, reducing the MLI:PC inhibition at neighboring Control PCs.

Minor comments:1. In the first sentence of the results, should "Figure 1A, B" be "Figure C, D"?

Yes, corrected.

1. The top of page 6 states "the mean mIPSC amplitude was ~10% greater in PC KL KO than in control", this does not appear to be the case in Figure 3E. control and KL KO look very similar here.

In this portion of the text citing the modest 10% increase in mIPSC amplitude, we are referring to the average amplitude of all individual mIPSC events in the PC KL KO condition; in the figure referred to by the reviewer (3E), we are instead referring to the average of all mIPSC event amplitudes per KL KO PC. Because of the dramatic difference in sample size for individual events vs cells, this modest difference rises to statistical, if not biological, significance. We include this individual event analysis only to suggest that, since we in fact saw a slightly higher event amplitude in the KL KO condition, it is unlikely that a reduced amplitude would have been a technical reason that we detected a lower event frequency.

1. Figure 3 D, duration, y-axis should be labelled "ms"

Event duration is no longer graphed or referenced. This has been replaced with total inhibitory charge.

**Reviewer #2 (Recommendations For The Authors):**
Methods:Pax2-Cre line: embryonal Cre lines sometimes suffer from germline recombination. Was this evaluated, and if yes, how?

The global loss of Kit signaling is incompatible with life, as seen from perinatal lethality in other Kit Ligand or Kit mutant mouse lines or other conditional approaches. Furthermore, a loss of Kit signaling in germ cells impedes fertility. Thus, while not explicitly ruled out, since conditional Pax2 Cre mediated Kit KO animals were born, survived, and produced offspring in normal ratios, we do not suspect that germline recombination was a major issue in this specific study.

Include rationale for using different virus types in different studies (AAV vs. Lenti).

This rationale is now included and reflects the intention to achieve infection sparsity in the smaller and less dense tissue of perinatal mouse brains.

How, if at all, was blinding performed for histological and electrophysiological experiments?

It was not possible for electrophysiology to be conducted blinded for the Kit KO experiments, owing to the subjects’ hypopigmentation. However, whenever feasible, resultant microscopy images or electrophysiological data sets were analyzed by Transnetyx Animal ID, and the genotypes unmasked after analysis.

Provide justification for limiting electrophysiology recordings to lobule IV/V and why MLIs in the middle third of the molecular layer were prioritized when inhibition of PCs is dominated by large IPSCs from basket cells. Why were 2 different internals used for recording IPSCs and EPSCs in PCs and MLIs? While that choice is justified for action potential recordings, it provides poor voltage control in PC voltage clamp. Both IPSCs and EPSCs could have been isolated pharmacologically using a CsCl internal.

The rationale for regional focus has been added to the text. For MLI action potential recordings, we opted to sample the middle third of the molecular layer so that we would not be completely biased to either classic distal stellate vs proximal basket subtypes. It is our hope, in future optogenetic interrogations, to simultaneously record the dynamics of all MLI subtypes in a more unbiased way. With respect to internal solutions, we initially utilized a cesium chloride internal to maximize our ability to resolve differences in GABAA mediated currents, which was the hypothesis-driven focus of our study. While we agree that utilizing a single internal and changing the voltage clamp to arrive at per-cell analysis of Excitatory/Inhibitory input would have been most informative, our decision to utilize pharmacological methods was driven by our experience that achieving adequate voltage clamp across large Purkinje cells was often problematic, particularly in adult animals.

Introduction:In the introduction, the authors state that inactivating Kit contributes to neurological dysfunction - their examples highlight neurological, psychiatric, and neurodevelopmental conditions.

The language has been changed.

General:Using violin plots illustrates the data distribution better than bar graphs/SEM.

We have included violin plots throughout, and we have changed p values to numeric values, both in the interest of presenting the totality of the data more clearly.

Synapses 'onto' PCs sounds more common than 'upon' PCs.

We have changed the wording throughout.

Figure 1:1F - there seems to be an antero-posterior gradient of Kit expression.

Though not explicitly pursued in the manuscript, it is possible that such a gradient may reflect differences in the timing of the genesis and maturation of the cerebellum along the AP axis. Regional variability is however now briefly addressed as a motivator for focused studies within lobules IV/V.

E doesn't show male/female ratios but only hypopigmentation.

This language has been corrected.

Figure 2 and associated supplementary figures:2A/B: The frequency of sIPSCs is very high in PCs, making the detection of single events challenging. How was this accomplished? Please add strategy to the methods.

We have added methodological detail for electrophysiology analysis.

How were multi-peak events detected and analyzed? 'Duration' is not specified - do the authors refer to kinetics? If so, report rise and decay. It is likely impossible to show individual aligned sIPSCs with averages superimposed, given that sIPSCs strongly overlap. Alternatively, since no clear baseline can be determined in between events, and therefore frequency, amplitude, and kinetics quantification is near-impossible, consider plotting inhibitory charge.

Given the heterogeneity of events, we now do not refer to individual event kinetics. As suggested, we have now included an analysis of the total inhibitory charge transferred by all events during the recording epoch.

S2: Specify how density, distribution, and ML thickness were determined in methods. How many animals/cells/lobules?

For consistency with viral injections and electrophysiology, the immunohistochemical analysis was restricted to lobule IV/V. This is clearer in the revision and detail is added in the methods.

S3:S3B: the labels of Capacitance and Input resistance are switched.

This has been corrected.

How were these parameters determined? Add to methods.

Added

In the previous figure the authors refer to 'frequency', in this figure to 'rate' - make consistent

This has been corrected.

D: example does not seem representative. Add amplitude of current pulse underneath traces.

We added new traces from nearer the group means and we now include the current trace.

F/G example traces (aligned individual events + average) are necessary.

We added example traces near the relevant group means for each condition.

Statement based on evoked IPCSs that 'synapses function normally' is a bit sweeping and can only be fully justified with paired recordings. Closer to the data would be the release probability of individual synapses is similar between control and Kit KO.

Paired recordings in both Kit Ligand and Kit receptor conditional knockout conditions is indeed an informative aim of future studies should support permit. For now, we have clarified the language to be more in line with the reviewer’s welcome suggestion.

S4:Histological strategy cannot unambiguously distinguish MLI-PC and PC-PC synapses. Consider adding this confound to the text.

We have added this confound to the discussion.

The observation that the pinceau is decreased in size could have important implications for ephaptic coupling of MLI and PC and could be mentioned.

We agree and have added this notion to the discussion.

Y-label is missing in B.

Corrected.

Figure 3 and associated supplementary figures:In the text, change PC-Cre to L7-Cre or Pcp2-Cre.

Changed

How do the authors explain a reduction in frequency, amplitude, and duration of sIPSCs in the KL KO but not in the Kit KO? Add to the discussion

We now address this apparent discordance in the discussion. Pax2 Cre mediates recombination weeks ahead of Pcp2 Cre. We therefore suspect that postnatal PC KL KO may be more phenotypic than embryonic MLI Kit KO because there is less time for developmental compensation. A future evaluation of the impact of postnatal Kit KO would be informative to this end.

As in Figure 2, plotting the charge might be more accurate.

We now plot total charge transfer.

Are the intrinsic properties in KL KO PCs altered? (Spontaneous firing, capacitance, input resistance).

We have added to the text that we found no difference in capacitance or input resistance between Purkinje cells from KL floxed homozygous Control animals versus those from KL floxed homozygous, PCP2 Cre positive KL KO animals. We plan to characterize both basal and MLI modulated PC firing in a future manuscript, especially since Pcp2 Cre mediated KL KO seems more phenotypic than Pax2 Cre mediated Kit KO, we agree that this seems a better testbed for investigating differences in both the basal, and the MLI-mediated modulations in, PC firing.

3D-F - Example traces would be desirable (see above, analogous to Fig. 2).

More example traces have been added.

Figure 4: 'In vivo mixtures' sounds unusual. Consider revision (e.g., 'to sparsely delete KL').

Changed

The observation that control PC sIPSC frequency is lower in KL OX PCs than in sham is interesting. This observation would be consistent with overall inhibitory synapse density being preserved. This could be evaluated with immunohistochemistry. For how far away from the injection area does this observation hold true?

Because we have now analyzed and failed to find an overt (per animal average) change in synaptic puncta size or density in the whole animal Control vs PCP2 Cre mediated KL KO conditions, we do not have confidence that it is feasible to pursue this IHC strategy in the sparse viral-mediated KL KO or OX conditions. To the reviewer’s valid point however, we intend to probe the spatial extent/specificity of the sparse phenomenon when we are resourced to complement the KL/Kit manipulations with transgenic methods for evaluating MLI-PC synapses specifically, potentially by GRASP or related methods that would not be confounded by PC-PC synapses. Transgenic MLI access would also facilitate determining the spatial extent to which opto-genetically activated MLIs evoke equivalent responses in Control vs KL manipulated PCs.

Y-legend in D clipped.

Corrected

Existing literature suggests that MLI inhibition regulates the regularity of PC firing - this could be tested in Kit and KL mutants.

For now, based upon transgenic animal availability, we have now included an evaluation of PC firing in the (Pax2 Cre mediated) Kit KO condition. PC average firing frequency, mean ISI, and ISI CV2 were not significantly different across genotypes. A KS test of individual ISI durations for Control vs Kit KO did reveal a difference (p<0.0001). We have added a supplementary figure (S6) with this data. It is possible that in the more phenotypic PC KL KO condition that we may find a difference in these PC spiking patterns of PC firing, however, we are also eager to test in future studies whether postnatal KL or Kit KO impairs the ability of MLI activation to produce pauses or other alterations in PC firing or in PF-PC mediated plasticity.

**Reviewer #3 (Recommendations For The Authors):**
Reference to Figure 1A in the Results section is slightly inaccurate. Kit gene modifications are illustrated in Figures 1A, B. Where Figure 1A shows Kit distribution. Please rephrase. Relatedly, the reference to Figs 1B - D are shifted in the results section, and 1E is skipped.

We have changed the text.

Please show cumulative histograms for frequency too for consistency with amplitude (e.g. Fig 2).

We have instead, for reasons outlined by other reviewers, documented total charge transfer for both Kit KO and KL KO experiments where sIPSC events were analyzed.

Fig S3: include example traces of PPR.

This is now included.

Include quantifications of GABAergic synapse density in Fig S4.

This is now included.

Include inset examples of KO in Fig S4A.

This is now included.

Add average puncta size graphs along Figure S4B. The effect apparent in the histogram of S4B is small and statistics using individual puncta as n values (in the 20,000s) therefore misleading.

Per animal analysis is now instead included in the figure and text.

Figure S4B y axis label blocked.

Corrected

Include quantification referenced in "As PSD95 immunoreactivity faithfully follows multiple markers of pinceaux size 40, we quantified PSD95 immunoreactive pinceau area and determined that pinceaux area was decreased by ~50% in Kit KO (n 26 Control vs 43 Kit KO, p<0.0001, two-tailed t-test)."

We added a graph of per animal averages, instead of in text individual pinceau areas.

Include antibody dilutions in the methods.

Added.

It's unclear from the text where the Mirow lab code comes from.

Detail has now been added in text.

Typo in methods "The Kit tm1c alle was bred...".

Corrected

Typo in Figure S4 legend "POSD-95 immuno-reactivity".

Corrected